# Coagulation using organic carbonates opens up a sustainable route towards regenerated cellulose films

Mai N. Nguyen [1,2,3], Udo Kragl [1,3,4], Ingo Barke [5], Regina Lange[5], Henrik Lund [4], Marcus Frank [1,6], Armin Springer[6], Victoria Aladin[1,3], Björn Corzilius [1,3] & Dirk Hollmann [1,3 ✉]

Due to their biodegradability, biocompatibility and sustainable nature, regenerated cellulose (RC) films are of enormous relevance for green applications including medicinal, environmental and separation technologies. However, the processes used so far are very hazardous to the environment and health. Here, we disclose a simple, fast, environmentally friendly, nontoxic and cost-effective processing method for preparing RC films. High quality non-transparent and transparent RC films and powders can be produced by dissolution with tetrabutylphosphonium hydroxide [TBPH]/[TBP]$^+$[OH]$^-$ followed by coagulation with organic carbonates. Investigations on the coagulation mechanism revealed an extremely fast reaction between the carbonates and the hydroxide ions. The high-quality powders and films were fully characterized with respect to structure, surface morphology, permeation and selectivity. This method represents a future-oriented green alternative to known industrial processes.

[1] Department Life, Light & Matter, Faculty for Interdisciplinary Research, University of Rostock, Albert-Einstein-Straße 25, 18059 Rostock, Germany. [2] School of Chemical Engineering, Hanoi University of Science and Technology, No. 1 Dai Co Viet street, 10000 Hanoi, Vietnam. [3] Department of Chemistry, University of Rostock, Albert-Einstein-Straße 3A, 18059 Rostock, Germany. [4] Leibniz Institute for Catalysis, Albert-Einstein-Straße 29a, 18059 Rostock, Germany. [5] Institute of Physics, University of Rostock, Albert-Einstein-Straße 23-24, 18059 Rostock, Germany. [6] Medical Biology and Electron Microscopy Centre, University Medicine Rostock, 18059 Rostock, Germany. ✉email: dirk.hollmann@uni-rostock.de

Albert Einstein once wrote: "We shall require a substantially new manner of thinking if mankind is to survive". The progressive industrialization and population growth are leading to continuous climate change and increasing environmental pollution throughout our planet. Therefore, we must ask ourselves: how can we tackle this great challenge for future generations? How can we change, or better, how can we prevent it? International initiatives for action have been developed by the United Nations under the title "Sustainable Development Goals". The sustainable development of the chemical science[1]—pioneered by Trost[2], Sheldon[3], outlined through the 12 principles of green chemistry proposed by Anastas and Warner[4,5], and further developed in respect to circular economy[6]—have changed the mindset of scientists. Thanks to the use of petrochemicals, most of Earth's pollution consists of non-degradable materials that generally require hundreds of years (or more) to decompose. However, the foreseen depletion of fossil reserves and the large impact of non-degradable materials on the environment and human health, e.g. (micro) plastics, are a strong motivation to use abundant sustainable resources. Therefore, the development of materials based on water, carbon dioxide, and biomass are of essential interest. In fact, biomass is a product of water and carbon dioxide produced by plants and consisting of cellulose (glucose units), hemicellulose, and lignin. Every plant, every natural bioresource and, therefore, also biowaste consists of these main components, however, in different proportions. Many microorganisms use cellulose for nutrition. Fungi or bacteria can cause the breakdown of the cellulose chains through enzymatic or radicals processes.

Due to its biodegradability and biocompatibility, cellulose, its modifications and derivatives are intensively used in numerous fields such as textile, packaging, paper production, environment, medicine, and separation technologies[7,8]. In most of these applications, cellulose is present in the form of cellulose II (so-called "regenerated" cellulose (RC)) that is the most thermodynamically stable modification[9]. It is accessible from crystalline cellulose (cellulose I) that is identical in its chemical structure, although it has different crystallinity due to changes in its hydrogen-bonding pattern[10].

An important challenge of both industrial and non-industrial processing of cellulose is its prior dissolution to obtain RC.

Current processes are quite problematic as they involve toxic solvents and reagents and involve several derivatization steps. For example, the Viscose process (production of Cellophane using alkaline media combined with carbon disulfide)[11] and the Cuprammonium route (production of Cuprophan using copper complexes with ammonium salts) for the production of regenerated cellulose fibers and films have dominated cellulose processing for more than a century[12]. However, these processes are accompanied by significant environmental pollution and therefore pose several economic and environmental challenges[13]. Thus, the environmental and social responsibility of companies along with pressure from customers prompted the search for less polluting, simplified and energy-efficient methods.

In recent years, RC have been successfully prepared by dissolution without derivatization in solvents such as LiCl/dimethylacetamide (DMAc)[14–16], alkali/urea, or alkali/thiourea[17–19] and ionic liquids (1-ethyl-3-methyl imidazolium acetate (EMIM Acetate))[20,21]. Particularly, amides are excellent solvents due to their high polarity. However, the amide functionality often implies reproductive toxicity[22,23]. Further challenges arise especially with ionic liquids such as EMIM acetate. Here, impurities and even traces of water are not tolerated. Moreover, water-containing solvent mixtures including N-methylmorpholine-N-oxide (Lyocell process)[24,25] or electrolyte solutions such as tetrabutylammonium hydroxide and tetrabutylphosphonium hydroxide (TBPH)[26] have been employed in the past.

Thus, our goal was to develop a simple, fast process at room temperature using nontoxic[27], recyclable, water-compatible, and renewable chemicals to produce RC. In this contribution we describe the preparation of regenerated cellulose by direct dissolution of cellulose (1) in aqueous TBPH (2) and the subsequent addition of organic carbonates, the so-called MDCell Process (named after both inventors). With this method high-purity RC powders and films are fabricated (Fig. 1). A short video showing the procedure is found in the electronic Supplementary information (Supplementary Movie 1). Organic carbonates such as propylene carbonate (3) are employed as promotor in the coagulation step, which requires seconds and has not been reported so far. Thus, in two simple and fast steps, MCC can be transformed into highly pure cellulose films under mild conditions. This developed method is the subject of a patent application[28].

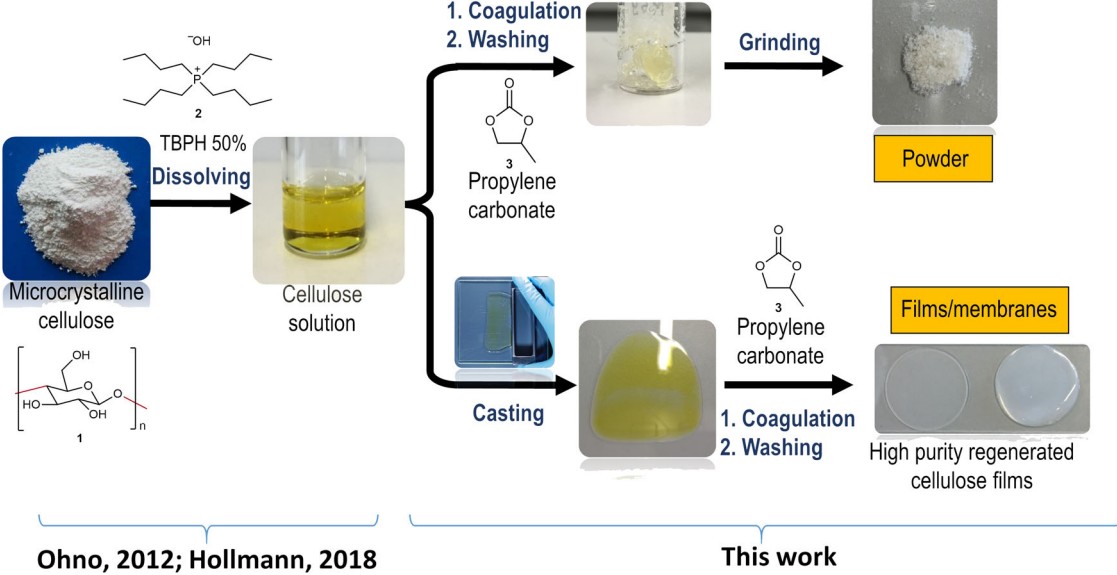

**Fig. 1 Preparation of regenerated cellulose using organic carbonates.** The combination of a previously reported method for cellulose dissolution with coagulation induced by organic carbonates provides a simple route to regenerated cellulose powers, films and membranes.

Importantly, the use of organic carbonates renders the process selective. Other reagents or solvents cause a reduction of the viscosity[29] or precipitation as RC powder as previously reported by a patent of Shimamoto and Ohno[30]. However, the coagulation with alcohols reported in the latter patent and the coagulation with organic carbonates described in this report involve two different mechanisms. A detailed investigation of this matter is described below.

Furthermore, the addition of co-solvents during dissolution with TBPH, enables the production of transparent membranes. The produced materials are characterized by Fourier transform Infrared spectroscopy (FT-IR), powder X-ray diffraction (XRD), and solid-state dynamic nuclear polarization-enhanced nuclear magnetic resonance spectroscopy (DNP-NMR). The surface properties of the synthesized films are investigated by scanning electron microscopy (SEM), transmission electron microscopy (TEM), and atomic force microscopy (AFM). By studying the permeation and selectivity of different dyes, the potential of the cellulose films in micro-, and ultrafiltration for waste treatment in a textile plant is evaluated.

## Results

**Formation of regenerated cellulose.** A water/TBPH mixture (50 wt.% of water), first described by Abe et al.[26] and later investigated by our group[29], is a highly efficient solvent for the dissolution of cellulose at levels of as much as 20 wt.%. TBPH constitutes a nontoxic[27] and recyclable solvent that can tolerate even large amounts of water and other standard solvents[29] without losing dissolution capacity. However, the search for nontoxic and renewable co-solvents to decrease the viscosity of the TBPH-cellulose solution and, thus, to increase the processability has continued. Surprisingly, organic carbonates such as propylene carbonate—obtainable by reaction of $CO_2$ with diols or epoxides[31]—do not decrease the viscosity. Instead of dissolution, spontaneous solidification occurs within seconds (Fig. 1). This solidification also proceeds in the presence of DMSO as co-solvent. The coagulated substances can be easily washed with water and dried at 80 °C to obtain granulates or powder. This powder is referred to as RC powder from here onwards.

Initially, we suggested a chemical modification of cellulose during the transformation with propylene carbonate (e.g., via ring-opening of **3** with the OH groups of cellulose) to form propylene carbonate–cellulose composites[32]. However, further investigations on the chemical and structural properties of these materials by means of FT-IR, XRD and $^{13}$C-NMR spectroscopy (Supplementary Fig. 1) ruled out any chemical transformation.

Propylene carbonate was not grafted onto the cellulose polymeric chains (FT-IR, Supplementary Fig. 1a). Thus, the dissolved cellulose coagulates without changing its chemical structure, indicating the "regeneration" process of the cellulose. Furthermore, analysis by $^{13}$C-NMR spectroscopy showed no changes in the chemical structure of C1–C6 of cellulose (Supplementary Fig. 1b)[33]. An analysis of the granules by XRD revealed transformation of the macroscopic structure compared to crystalline cellulose. Reflections at 12.1° and 20.6° indicate the formation of cellulose II (regenerated cellulose) (Supplementary Fig. 1c). Interestingly, the $^{13}$C-NMR spectra highlight the purification effect of the process and the improved quality of the regenerated cellulose (Supplementary Figs. 2 and 3). The impurities found in crystalline cellulose (sharp signals of glucose monomers and dimers of cellobiose) are not present in the $^{13}$C-NMR of re-dissolved RC powder.

As an alternative to the **3** employed, other organic carbonates can also be used to obtain regenerated cellulose (Supplementary Fig. 4). The use of vinyl ethylene carbonate, butyl carbonate, and ethylene carbonate leads to the same regenerated cellulose. However, propylene carbonate constitutes the best alternative due to price, availability, and low melting point. It is remarkable that the amount of cellulose (Supplementary Fig. 5a), the amount of propylene carbonate (Supplementary Fig. 5b), and the origin of cellulose (Supplementary Fig. 5c) do not influence the structure of the regenerated cellulose.

**Investigation of the coagulation mechanism.** Since no reaction of **3** with cellulose has been detected, we were interested in the nature and mechanism of the coagulation–regeneration process triggered by **3**. Therefore, TBPH-cellulose mixture was

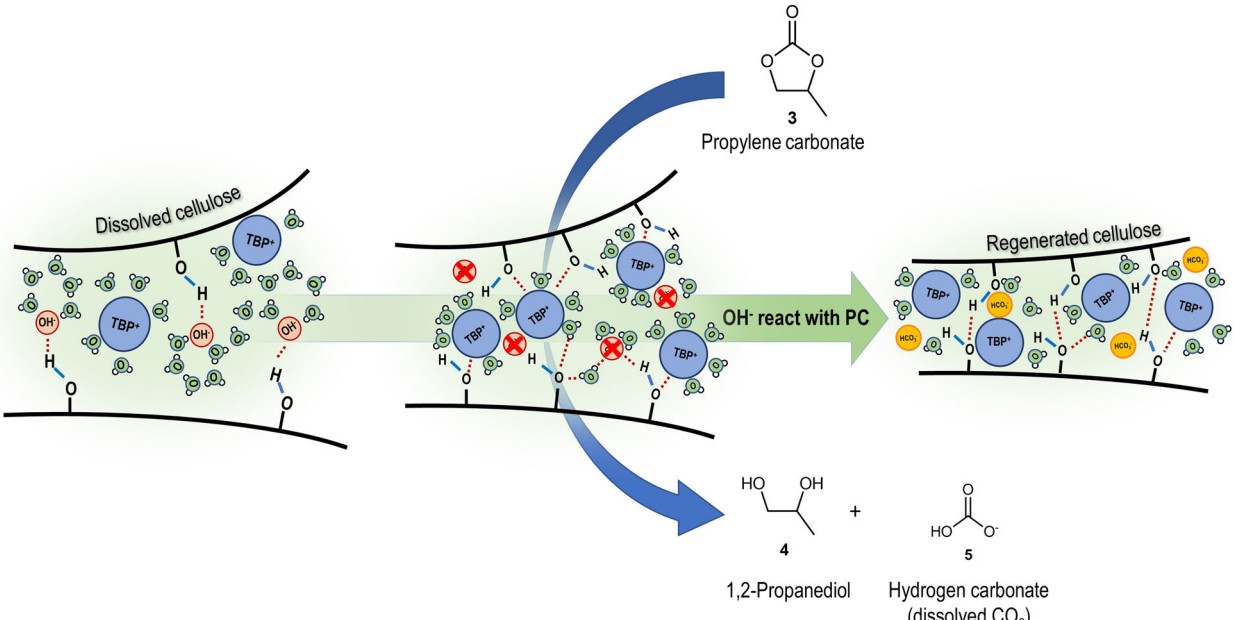

**Fig. 2 Proposed mechanism for the precipitation of cellulose triggered by propylene carbonate.** The hydroxide ions [OH]⁻ of TBPH can break the strong hydrogen bonds between the cellulose chains, which leads to the dissolution of cellulose. However, if **3** is present, a nucleophilic attack of the [OH]⁻ ions occurs. The reaction generates 1,2-propanediol (**4**) and hydrogen carbonate (**5**). Thus, no hydroxide ions are available to stabilize the dissolved cellulose and the hydrogen bonds are rearranged between the cellulose chains.

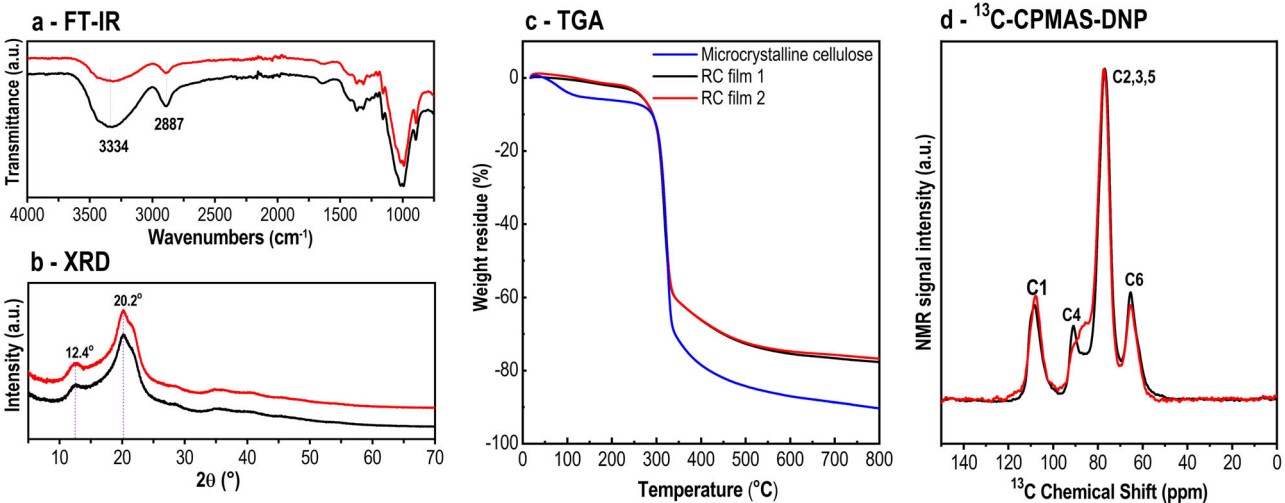

**Fig. 3 Characterization of RC films 1 (black) and 2 (red). a** FT-IR spectra. **b** XRD patterns. **c** Thermal gravimetric analysis. **d** $^1H$–$^{13}C$ cross polarization (CP) MAS NMR spectra under DNP at 110 K. Spectra have been scaled to the same signal intensity.

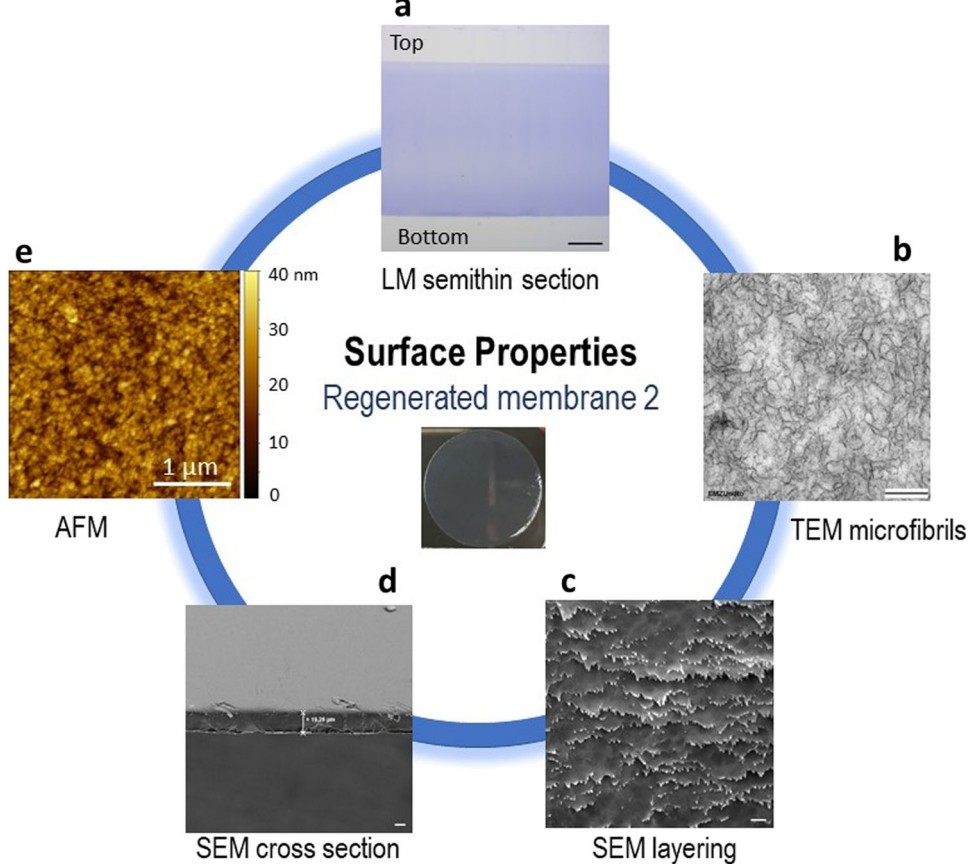

**Fig. 4 Surface properties of RC film 2. a** Light microscopic (LM) image of a semithin cross section of a RC film processed and embedded for electron microscopy section stained with Toluidine blue; scale bar is 50 μm. **b** High resolution TEM imaging reveals the ultrastructure of the RC films with multiple branched cellulose fibrils in a thin section (~50 nm) contrasted with lead citrate and uranyl acetate; scale bar is 200 nm. **c** SEM micrograph of a dried membrane opened out by cross-braking under liquid nitrogen which exhibits multiple uniform layers of the RC film structure; scale bar is 200 nm. **d** SEM image of a cut through the dried membrane showing the thickness of the dried RC film; scale bar is 10 μm. **e** AFM image of the dried film providing the primary roughness.

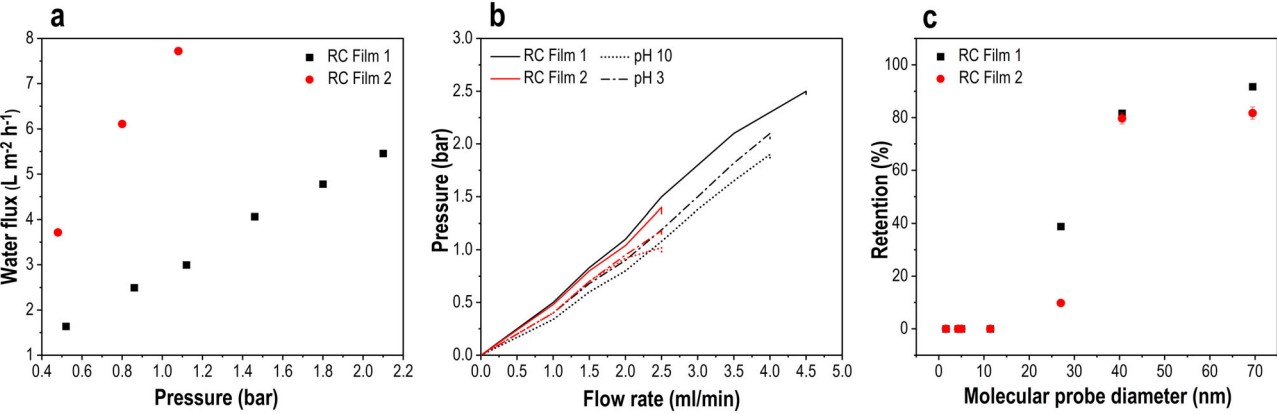

**Fig. 5 Membrane properties of RC films 1 and 2. a** Water flux. **b** Stability test before and after storage at different pH for 2 days. **c** PEG retention of the films at a PEG concentration 2000 ppm (1.5 ml min⁻¹ flow rate, 1 bar). Error bars in **c** correspond to the standard error.

investigated by $^{13}$C-NMR spectroscopy after coagulation with **3** (Supplementary Fig. 6). No changes at the tetrabutylphosphonium ion [TBP]$^+$ were observed. However, new signals at 160.4, 68.4, and 67.8 ppm appeared. Thus, a detailed study of the mixture of TBPH and **3** was conducted by means of $^1$H, $^{13}$C, DEPT, HSQC, HMBC, COSY NMR experiments (see Supplementary Figs. 7–14) suggesting a reaction between **3** and the hydroxide anion [OH]$^-$. These signals can be assigned to the [HCO$_3$]$^-$ anion (**5**, dissolved CO$_2$), as well as to 1,2-propanediol (propylene glycol, **4**). Indeed the [OH]$^-$ ion plays an essential role in the dissolution abilities of the TBPH solution[29]. The [OH]$^-$ ions are converted selectively via the reaction with organic carbonates, thus the coagulation of the cellulose occurs immediately. The proposed mechanism for the coagulation toward regenerated cellulose is illustrated in Fig. 2.

**Formation and structural characterization of RC films**. To investigate the potential of our new method, we employed it for the production of films (Fig. 1, Supplementary Movie 1). A typical procedure involves the dissolution of 20 wt.% cellulose in TBPH 50 wt.% at room temperature (23 °C) for 30 min to ensure the complete solubility of cellulose. The resulting clear solution was cast onto a glass plate to produce a cellulose solution layer of a defined thickness (e.g., 500 μm)[34]. The casted liquid was quickly immersed into a propylene carbonate bath for coagulation. Note, that no change of the optical properties occurs. The regenerated cellulose films can be washed with distilled water to remove all water-soluble reagents and impurities.

Distinct effects were observed using DMSO as co-solvent. It is well known that DMSO is an excellent promotor in dissolution processes[35], reducing the required solvent amount[36], and reducing the viscosity of the solution without precipitation of cellulose[29]. In this study, the quality of the films is remarkably improved by DMSO. They become more flexible, homogeneous, and almost fully transparent. Thus, we investigated two films produced from 20 wt.% cellulose in TBPH 50 wt.%, RC film 1 without DMSO, and RC film 2 with DMSO. For the latter, the ratio between DMSO and TBPH was 1:1.

Next, we wanted to decipher the chemical and mechanical properties of both films. The chemical composition (by IR spectroscopy) and the powder diffraction pattern (by XRD) of both cellulose films were not different from the powder mentioned before (Fig. 3a, b). The same was observed by thermogravimetric analyses (TGA), which verify the thermal stability before and after cellulose regeneration, as shown in Fig. 3c. Both films are stable up to 250 °C. Prior to this temperature, the weight loss is mainly due to the volatilization

and vaporization of moisture from the cellulosic samples[37]. In fact, the regenerated cellulose films exhibited only a small continuous weight loss compared to the crystalline sample. The pyrolysis processes, which ranged from 250 to 350 °C (onset decomposition temperature 271 °C), did not shift to a higher or lower temperature, indicating no change in the thermal stability of the regenerated cellulose.

To shed further light on structural differences between RC film 1 (non-transparent) and RC film 2 (transparent) we performed $^{13}$C-NMR experiments in the solid-state under magic-angle spinning (MAS) and DNP on frozen, wet films (Fig. 3d, Supplementary Fig. 15). The results show significant differences in the spectral distribution of the glucose-linking residues C1 and C4. A significant fraction of the residue C4 of RC film 2 is shifted upfield, away from the resonance signal of bulk crystalline cellulose indicating a larger number of surface-accessible sites[38,39]. In contrast, RC film 1 shows a major contribution from crystalline cellulose. Solid-state NMR measurements of wet films as well as of dried samples under ambient conditions confirm this tendency (Supplementary Fig. 16). These variations in surface-to-bulk ratio are correlated with the transparency of RC film 2 and opacity of RC film 1.

The detection of amorphous, surface-accessible sites was a strong motivation for the investigation of the surface morphology of the films. Applying light microscopy (LM), TEM, SEM, and AFM, the structure and morphology of the films was investigated. In Fig. 4, a summary of the surface characterization techniques for the most important RC film 2 is shown. Details for RC film 1 and a comparison with RC film 2, further information on the conditions as well as the images are given in Supplementary Figs. 17–19.

First, the wet RC films were inspected by LM with toluidine blue (TB, Fig. 4a). Here, a homogeneous distribution of the TB over the entire size of regenerated cellulose was detected. The thickness of the films decreased to around 200 μm, which is nearly the same high as the casting high of 500 μm. The reduction in thickness was probably caused by the coagulation process and partially drying during the LM preparation process. No capillary pores were visible. Only small differences in the density and roughness of the front and back side of the film can be observed which are probably due to the casting on the glass surface. Interestingly, contrast reagents such as lead citrate and uranyl acetate can visualize the cellulose microfibrils in subsequent high resolution TEM imaging[40] (Fig. 4b). Here, an irregular, net-like structure of branched microfibrils was observed, with a size of the net meshes in the range of ~30–100 nm. Furthermore, the film surface appears relatively smooth by TEM inspection at the border to the resin (Supplementary Fig. 17a, d).

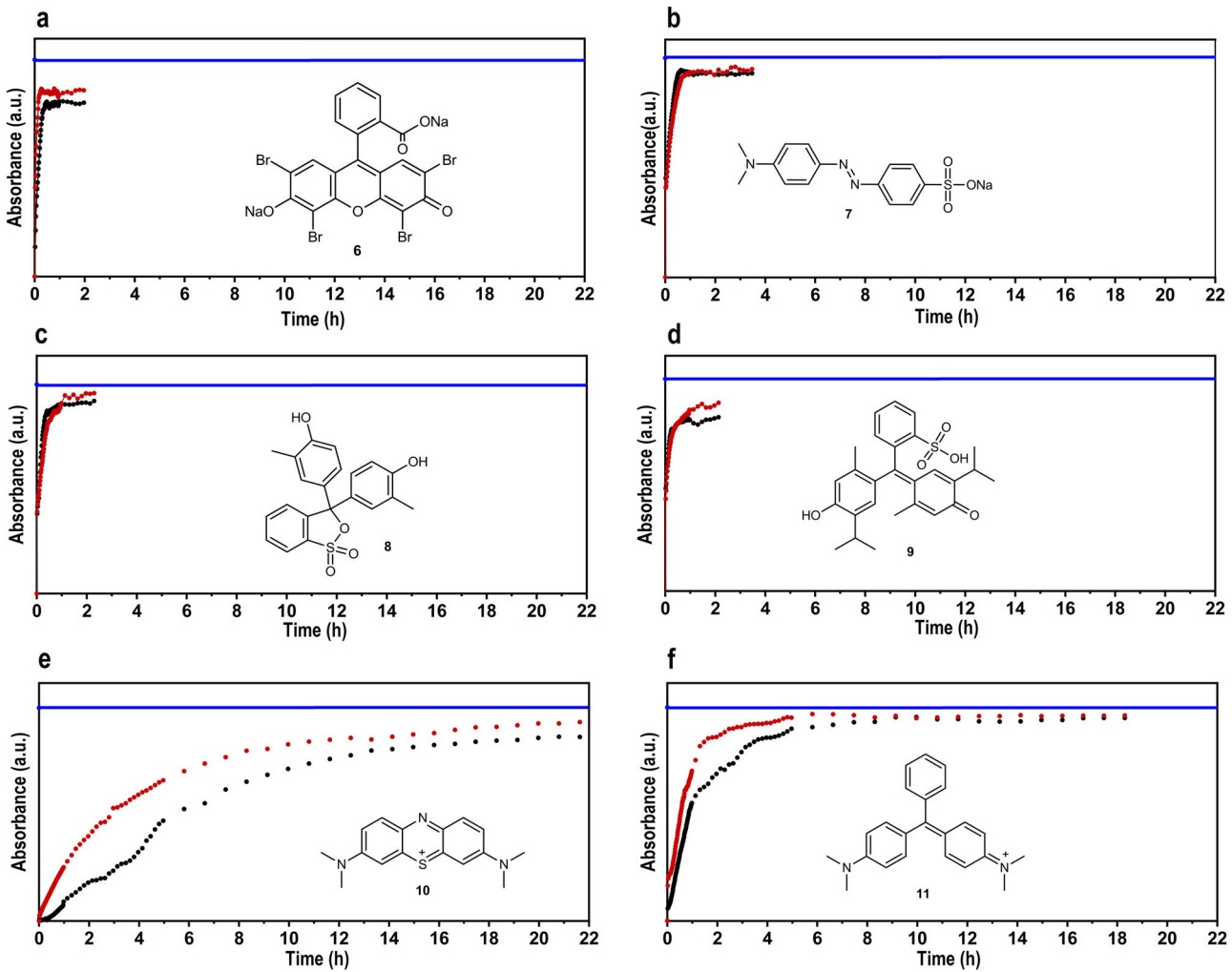

**Fig. 6 Online continuous flow UV–Vis spectroscopy for charged dyes. a** Eosin Y disodium salt (**6**), **b** Methylene orange (**7**), **c** Cresol red (**8**), **d** Thymol blue (**9**), **e** Methylene blue (**10**), **f** Malachite Green (**11**). Blue line: Start solution ($c = 10$ ppm). Black dots: absorbance after going through RC film 1. Red dots: absorbance after going through RC film 2.

Further drying in air enables the application of SEM and AFM. Under SEM conditions, an almost perfect surface with a compact, very uniform, dense, but layered structure was detected (Fig. 4c). Again, no capillary pores were observed. These layers correspond well to the cellulose microfibrils seen in TEM. Interestingly, the film thickness of RC film 1 (without DMSO) decreases to 33 μm and even to 20 μm of RC film 2 (with DMSO) (Fig. 4d). To measure the surface roughness, the dry films were investigated by AFM (Fig. 4e and Supplementary Fig. 19). These experiments revealed a very smooth surface with low average surface roughness of only $R_a \approx 3$ nm. All results indicate a fast and homogeneous penetration and reaction with the organic carbonate **3**.

**Utilization of the RC films**. After investigating the structure, we were keen to utilize both films as membranes. First, the water flux of both synthesized films was investigated (Fig. 5a). In general, the water flux is affected by the porosity, stability, and the distribution and size of the pores in the membrane[41]. Thus, the water flux at several differential pressures was measured showing an increase between 1.63 (at 0.52 bar) to 5.45 (at 2.1 bar) L h$^{-1}$ m$^{-2}$ for RC film 1. Interestingly, a higher water flux for RC film 2 was detected. The resulting water flux raised to 7.72 L h$^{-1}$ m$^{-2}$ at 1.08 bar over pressure. The resulting permeability (water flux divided by pressure) was 2.8 and 7.5 L m$^{-2}$ h$^{-1}$ bar$^{-1}$ for RC films 1 and 2, respectively.

In addition, the mechanical durability based on the hydraulic pressure of the flow through the film was investigated in detail (Fig. 5b). This also results in the limitation of the films. Indeed, the RC films 1 (prepared without DMSO) and 2 (prepared with DMSO) show a different behavior which are in agreement with the water flux measurements. RC film 1 was destroyed at a pressure of 2.5 bar corresponding to the maximum flow rate of 4.5 ml min$^{-1}$. Meanwhile, RC film 2 was broken even at a pressure of 1.4 bar corresponding to a maximum flow rate of 2.5 ml min$^{-1}$. These results indicate that RC film 1 is more stable than RC film 2. This forced us to investigate the long-term stability even at different pH. Thus, the films were immersed in acid media (pH = 3) and basic media (pH = 10) for 2 days and measured. The hydraulic pressure is comparable with the fresh RC films, showing a high storage stability in these media.

The identification of the pore size of both films was carried out using polyethylene glycol (PEG) as molecular probes (Fig. 5c). The molecular weight of PEG (from 1000 to 1,000,000 g mol$^{-1}$) is correlated to its Stokes–Einstein diameter (from 1.57 to 69.46 nm)[42]. In fact, both films have nearly the same pore distribution of 10–70 nm. Since no tube-like pores are presented, we believe that the filtration ability is restricted by the interlayer space of the microfibrils detected by TEM and SEM. The best retention was achieved with PEO 1,000,000 (polyethylene oxide, molecular probe diameter of 69.46 nm). The rejection of 92% (RC film 1) and 82% (RC film 2),

respectively, identify the molecular weight cut-off for both films. This opens up the application of these films as micro- and ultrafiltration membranes.

The material properties reveal a strong dependence on the presence of DMSO during the coagulation process. First, DMSO reduces the surface tension[29], which allows a fast diffusion of **3** and thus a faster coagulation. This can lead to a larger distance between the fibrillary structures in this material[43], indicated by the three-fold higher permeability as well as larger pore distribution compared to RC film 1. The weaker interaction and higher porosity are responsible for the lower stability.

Finally, we demonstrate the application of our RC films as membranes for the removal or separation of dyes. We selected six organic dyes with different charge, which are classified with negative charges (eosin Y disodium salt—**6**, methylene orange—**7**), no (neutral) charges (cresol red—**8**, thymol blue—**9**) and positive charges (methylene blue—**10**, malachite green—**11**). Using continuous flow online Ultraviolet-visible spectra (UV–Vis) spectroscopy (see Supplementary information), we were able to detect a charge selectivity of the regenerated cellulose films (Fig. 6). Negatively charged and neutral dyes (Fig. 6a–d) can penetrate without barriers through the cellulose microfibrils, resulting in an immediate change in absorptivity. The equilibrium was reached after 20–40 min. Surprisingly, positively charged dyes are retained and delayed by both RC films (Fig. 6e, f). Equilibrium was reached after more than 16 h. Taking into account that the interlayer distance of the microfibrils is much larger than the dye molecules, these results suggest a negative charge surface of the cellulose. This was proven by zeta potential measurements of the cellulose films ($\zeta = -20$ mV between pH 5 and 9). Indeed, the induced polarization at the membrane surface may correspond to a sorption of negative hydroxyl ions or to partially ionization via the positive charged $[TBP]^+$ cation.

## Discussion

Applying this methodology, regenerated cellulose powders and films of high quality can be obtained. The two-step procedure involves the dissolution of cellulose in TBPH followed by coagulation triggered by addition of organic carbonates. The optional application of a second solvent (besides water) results in the production of transparent films. This methodology is simple, fast, does not require the use of inert gas, can be performed at room temperature and is environmentally friendly. All the employed chemicals are nontoxic and can be in principle obtained by fixation of $CO_2$ (organic carbonate) or are recyclable ($[TBP]^+$). Water is tolerated in every step. Using NMR spectroscopy, the full mechanism was evaluated and clarified. Propylene carbonate reacts with the $[OH]^-$ anions to release 1,2-propanediol as well as hydrogen carbonate. This is complementary to the coagulation processes using anti-solvents. The $[TBP]^+$ cation is not affected and can be easily separated by washing. Therefore, it is accessible for recycling and reuse. Regarding this, we would like to mention some strategic aspects regarding the recycling of the materials used in our process. The most important substance is TBPH. Even though it is nontoxic to the environment, TBPH is expensive and—as a Brønsted base—corrosive. Since phosphorus is also a finite resource on our planet, recycling is essential. In principle, TBPH can be recycled via ion exchange as reported by Kilpeläinen[44]. The propylene carbonate coagulation bath can be used several times, even impurities of water, DMSO or TBPH do not result in the loss of the coagulation ability. The mixtures of **3/4** and TBPH could be separated via solvent extraction or distillation. This is subject of further research. The regenerated cellulose produced can be easily re-dissolved with TBPH, enabling an easy recycling.

With this method, very smooth, highly pure and dense films can be fabricated. The RC films have a dense layer structure of microfibrils with a mesh of about 10–70 nm. The RC films are stable in water and can be stored for several months. Recycling of the membrane can be done easily via another dissolution step in TBPH. However, the mechanical stability of the films is still low. Only up to 2–3 bar pressure could be applied to the film during the water flux measurements. Therefore, our future research will focus on the development of modified films and membranes to achieve better stability under wet and dry conditions as well as a distinct pore size distribution. Another focus will be the prevention of the shrinking process during drying as well as the introduction of higher flexibility in order to enable the application as a transparent film. These results indicate great potential of this methodology for the sustainable preparation and recycling of the cellulose powders, films, and membranes for ultra- or microfiltration.

## Methods

**Materials**. TBPH containing 40 wt.% in water was purchased from Acros and concentrated through rotary evaporation under 70 bar, 40 °C to get higher concentration (50 wt.%). Microcrystalline cellulose (MCC) with the size 20 μm used in this study was purchased from Sigma Aldrich. Propylene carbonate (**3**, 99.7% purity), vinyl ethylene carbonate (VEC, 99% purity), ethylene carbonate (EC, 98% purity), butylene carbonate (BC, 98% purity) were obtained from Sigma Aldrich. Dimethyl sulfoxide was supplied from Sigma Aldrich. PEG with molecular weight in the range as follows: PEG 950–1050, PEG 35,000, PEO 200,000, PEO 400,000, PEO 1,000,000 were provided from Sigma Aldrich. Methylene blue, malachite green, cresol red, thymol blue, eosin Y disodium salt and methylene orange were obtained from Sigma Aldrich.

**Sample preparation for DNP experiments**. Each film was soaked in 35 mM AMUPol [SATT Sud-Est, Marseille] $D_2O/H_2O$ (90/10 vol.-%) mixture for one hour[45]. Each film was then transferred into a 3.2 mm sapphire MAS sample rotor (Bruker) while in the wetted state and closed with a vessel cap before freezing inside the spectrometer.

**Fourier transform infrared spectroscopy**. Infrared spectra of samples were recorded with a Bruker FT-IR Alpha II spectrometer in ATR mode. The specimens were measured directly with a scan range from 400 to 4000 cm$^{-1}$.

**Liquid state NMR**. The samples were measured without using deuterated solvents with a Bruker spectrometer AVANCE Neo 500 at 125.8 MHz (10,000 accumulations, 2 s pulse delay, at 25 °C). The NMR spectra were calibrated to an internal standard/reference namely 1,4-dioxane.

**Solid-state NMR**. Solid-state NMR experiments were performed under MAS using a Bruker AVANCE III HD spectrometer operating at 400.2 MHz proton frequency with a Bruker ASCEND DNP 9.4 T widebore (89 mm) magnet and a MASWVT400W1 BL4 X/Y/H triple channel probe operating in double-resonance mode. Samples were spun in 4 mm $ZrO_2$ rotors (Bruker) at room temperature and 8 kHz MAS frequency. Radio frequency (rf) pulse powers were set to 83 kHz and 50 kHz for $^1$H and $^{13}$C, respectively. $^1$H power was matched to $^{13}$C during Hartmann–Hahn cross polarization (CP) with 1.5 ms contact time, SPINAL64 at 83 kHz was used for broadband decoupling of $^1$H during the detection windows of 20 ms with a recycle delay of 1 s. In total, 3072 and 6144 scans were accumulated for dried films 1 and 2, respectively. In total, 69,632 and 147,456 scans were accumulated for the wet film 1 and 2 samples, respectively. Spectra were processed in TopSpin 4.0.6 (Bruker) by exponential window multiplication (EM) with 50 Hz broadening parameter (LB) before FT and phase adjustment. The chemical shift was referenced to TMS (0 ppm) by using an external adamantane standard.

**NMR under DNP**. DNP-NMR experiments were performed under MAS using a commercially available Bruker AVANCE II DNP spectrometer operating at 400.2 MHz proton frequency with a Bruker Ultrashield 9.4 T widebore (89 mm) magnet. In total, 263.4 GHz microwaves were produced by a Bruker gyrotron with 60 mA beam current. Experiments were performed at 111 K with microwave irradiation and at 106 K without microwave irradiation; temperature was detected with a thermocouple inside the MAS stator. rf pulse powers were set to 100 kHz and 50 kHz for $^1$H and $^{13}$C, respectively. $^1$H power was matched to $^{13}$C during Hartmann–Hahn CP with 1.5 ms contact time, SPINAL64 at 100 kHz was used for broadband decoupling of $^1$H during the detection windows of 17 ms. MAS with a spinning frequency of 8 kHz was used. Polarization was allowed to build up for 3 s before each scan. Spectra were processed in TopSpin 4.0.6 (Bruker) by EM with

50 Hz broadening parameter (LB) before FT and phase adjustment. For the enhancement measurements, spectra with and without microwave irradiation were normalized to the equal number of scans (64 scans for spectra with and 768 (film 1) and 17,408 (film 2) scans without microwave irradiation). The intensity ratio of the signal with and without microwave irradiation is then given as the enhancement factor ε. The chemical shift was referenced to TMS (0 ppm) by using an external glycine standard.

**X-ray diffraction**. XRD powder pattern were recorded on a Panalytical X'Pert diffractometer equipped with a Xcelerator detector using automatic divergence slits and Cu kα1/α2 radiation (40 kV, 40 mA; λ = 0.15406 nm, 0.154443 nm). Cu beta-radiation was excluded using a nickel filter foil. The measurements were performed in 0.0167° steps and 100 s of data collecting time per step. The samples were mounted on silicon zero background holders. The obtained intensities were converted from automatic to fixed divergence slits (0.25°) for further analysis. Peak positions and profile were fitted with Pseudo-Voigt function using the HighScore Plus software package (Panalytical). Phase identification was done by using the PDF-2 database of the International Center of Diffraction Data.

**Thermal analysis**. The TGA were carried out by using a Setram Labsys thermal analyzer. Amounts of about 5–15 mg of the carefully dried samples were sealed in aluminum crucibles and studied in the temperature range of 20–800 °C with heating rates of 10 °C min$^{-1}$ under inert atmosphere (50 ml/min nitrogen).

**Atomic force microscopy**. AFM Topographies of dried cast and spin-coated samples have been obtained with a commercial device (Park Systems XE100) in dynamic mode using metal coated Si cantilevers (type ACTA, AppNano, Al-coating, 300 kHz and HA-HR, SpectrumInstruments, Au coating, 380 kHz). In order to avoid excessive sample contact, the device was operated in the so-called non-contact mode with small cantilever oscillation amplitudes of a few nonometer and moderate setpoint (~70% of free amplitude), while the dither frequency was kept constant (amplitude modulation). Multiple sample locations were probed to ensure validity of obtained morphology.

Spin coating (spin coater SPIN150i, SPS-Europe) was used to obtain a very smooth homogenous film especially for AFM measurements. Two hundered microliters of the cellulose solution without/with DMSO, respectively, was dispensed on a glass cover slip that was rotating at 10,000 rpm. After 5 min the rotation speed was decreased to 1000 rpm and **3** was added for 1 min.

**Scanning ELectron Microscopy**. Specimens were air-dried or freeze dried. For freeze drying specimens were rapidly frozen on a liquid nitrogen precooled polished copper block, immediately transferred to a liquid nitrogen precooled exsiccator and freeze dried under vacuum (5 × 10$^{-2}$ mbar) with subsequent warming to room temperature. Membrane fracturing was performed on rapidly frozen samples under liquid nitrogen with a pair of precooled forceps. Specimens were mounted on SEM stubs with adhesive carbon tape (Plano, Wetzlar, Germany) and coated with a carbon layer of ~15–20 nm (Leica SCD500, Leica Microsystems, Wetzlar Germany). Specimens were viewed in a field-emission SEM (Zeiss Merlin VP compact, Carl Zeiss Microscopy, Oberkochen, Germany) equipped with HE-SE and in-lens-Duo detectors. Images with a size of 1024 × 768 pixels were recorded at different steps of magnification. Measurements of distances were performed using the SmartSEM measurement tools (Carl Zeiss Microscopy).

**Transmission electron microscopy**. Specimens were cut-off wet membranes with razor blades to a size of ~1 × 2 mm for subsequent processing for TEM. After fixation with an aqueous solution of 1% osmium tetroxide for 1 h and washes in distilled H$_2$O, dehydration through an ascending acetone series to 100% acetone was followed by infiltration with epoxy resin (Peon 812, Serva, Heidelberg, Germany) starting in a 1:1 mixture of acetone and resin o/n, followed by pure resin for 4 h. Specimens were transferred to rubber molds and the resin was allowed to cure at 60 °C for 2 days. The membranes were exposed from the resin blocks with a trimming tool (Leica EM Trim 2, Leica Microsystems, Wetzlar, Germany). Semi-thin sections (~0.5 μm) and thin sections (~50–70 nm) were cut on an ultra-microtome (Ultracut S, Reichert, Wien, Austria) with a diamond knife (Diatome, Biel, Switzerland). Semithin sections were stained with an aqueous solution of TB to visualize specimens for further trimming prior to thin sectioning and ultra-structural inspection. Thin sections were transferred to copper grids and were either examined directly without further contrasting agents or were alternatively contrasted with uranyl acetate and lead citrate. The sections were examined on a Zeiss EM902 electron microscope operated at 80 kV (Carl Zeiss, Oberkochen, Germany). Digital images were acquired with a side-mounted 1x2k FT-CCD Camera (Proscan, Scheuring, Germany) using iTEM camera control and imaging software (Olympus Soft Imaging Solutions, Münster, Germany).

**Light microscopy**. Semithin sections stained with TB were examined with a LM (Zeiss Axioskop 40, Carl Zeiss, Göttingen, Germany) and digital images were recorded with a camera (Zeiss AxioCam ERc5s) using acquisition software with integrated measurement tools (Zeiss ZEN blue edition).

**Ultraviolet-visible spectra**. UV–Vis transmission spectra were recorded by a fiber system consisting of an AvaLight-DH-S-BAL Balanced Power Light Source and an AvaSpec-ULS2048 StarLine Versatile Fiber-optic Spectrometer (Avantes). Online detection of the dyes was performed using homemade filter equipment connected to a micro-flow cell (Avantes).

**Zeta potential**. Zeta potential measurements were performed on RC films (2 × 1 cm) using the SurPASS system (Anton Paar, Ostfildern, Germany) to gain information on the surface charge. The measurements were performed in a 0.001 mol/L KCl solution ranging from pH 5.0 to 9.0 with a gap height of 100 μm. The streaming current was determined depending on the pressure (max. 400 mbar). Finally, the zeta potential was calculated according to the method of Helmholtz–Smoluchowski.

## Data availability

The authors declare that the data supporting the findings of this study are available within the paper and its Supplementary Information files, or from the corresponding author upon request.

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

## Acknowledgements

The authors would like to thank Dr. Dirk Michalik and Heike Borgwaldt for NMR measurement and Dr. Renate Nareyka for TGA measurement. We would like to thank Dr. Christina Oppermann for HPLC introduction. We thank Ute Schulz (EMZ) for support with TEM embedding and sectioning. D.H. and M.N.N. would like to thank Marcus Müller, ITMZ Rostock for the support with the movie production as well as Dr. Henrike Rebl for the measurement of the Zeta potential. D.H. thanks Dr. Esteban Mejia and Prof. Björn Corzilius for checking the English language. This work has been supported by the RoHan Project funded by the German Academic Exchange Service (DAAD, No. 57315854) and the Federal Ministry for Economic Cooperation and Development (BMZ) inside the framework "SDG Bilateral Graduate school program". Access to DNP-NMR was provided by the Center for Biomolecular Magnetic Resonance (BMRZ), Frankfurt. Open access funding provided by Projekt DEAL.

## Author contributions

M.N.N. carried out the experiments, analyzed data, and wrote the manuscript. D.H. supervised the project, analyzed the data and wrote the manuscript. H.L. performed XRD measurements and analysis of data. AFM measurements and analysis were conducted by I.B. and R.L. M.F. and A.S. performed SEM and TEM measurements. V.A. and B.C. conducted the DNP measurements. U.K. corrected the manuscript. The paper was written through contributions of all authors. All authors have given approval to the final version of the paper.

## Competing interests

A patent application has been submitted by University of Rostock for this process, with M.N.N., D.H., and U.K. as inventors[28]. All other authors declare no competing interests.
