## [Peer Review File · Communications Chemistry]

Reviewers' comments:

Reviewer #1 (Remarks to the Author):

The authors reported a new methodology that regenerated cellulose powders and films can be obtained. The two-step procedure, which involves the dissolution of cellulose in TBPH followed by the coagulation triggered by propylene carbonate, is very simple and recyclable. This process will have a strong impact on cellulose research. The analysis is reasonable. I recommend publishing with minor revision. Please consider my comments as follows.

1. Mechanical properties

The authors succeeded to obtain flexible and homogenous cellulose films. Have you measured the mechanical properties (tensile strength, ultimate elongation, etc.)? If you add such information in the manuscript, it would be great for understanding deeply the effect of DMSO.

2. P13, L262

...(PEG) as molecular probes (Figure 6B).

I think Figure 5B is correct.

3. P14, L282

The authors mentioned that "these results suggest a negative charge surface of the cellulose." I understand the results, but why? The OH-group of cellulose is not modified with any chemicals in this process, right? Please add the explanations in the manuscript.

4. P15, L301

The [TBP]⁺ anion is not...

I think cation is correct.

5. P19, L399

Please add the apparatus information.

Reviewer #2 (Remarks to the Author):

Development of sustainable route for the clean production of regenerated cellulose is of vital importance for the wider application of cellulose and sustainable development of human society. In this work, the authors used TBPH solution to dissolve MCC and get regenerated cellulose by adding organic carbonates. Some properties of the regenerated cellulose film were characterized. The results obtained are quite interesting, but the following points have to be fully addressed before acceptance. So major revision is needed.

Detailed comments:

1. The title should be modified. At least, TBPH solution should be mentioned.

2. I think the properties of the regenerated cellulose film obtained from this work should be compared with the regenerated cellulose film obtained by Viscose process or other reported processes based on tests or literature results, for instance, mechanical strength, barrier properties?

3. Page 12, line 231, please indicate how the thickness of the films decreased to around 200 nm? By drying?

4. Page 19, line 400, please indicate the gas flow speed of nitrogen.

5. Please indicate the test mode of AFM.

Response to Reviewers

Reviewer #1 (Remarks to the Author):

The authors reported a new methodology that regenerated cellulose powders and films can be obtained. The two-step procedure, which involves the dissolution of cellulose in TBPH followed by the coagulation triggered by propylene carbonate, is very simple and recyclable. This process will have a strong impact on cellulose research. The analysis is reasonable. I recommend publishing with minor revision. Please consider my comments as follows.

1. Mechanical properties

The authors succeeded to obtain flexible and homogenous cellulose films. Have you measured the mechanical properties (tensile strength, ultimate elongation, etc.)? If you add such information in the manuscript, it would be great for understanding deeply the effect of DMSO.

Comment: We totally agree with the reviewer. Indeed, the mechanical properties are highly important for the future applications of our cellulose films. For our application in filtration, the mechanical durability based on the hydraulic pressure of the flow through the film is of the highest interest. Thanks to the comment of the reviewer, we performed these stability tests. The RC films 1 (prepared without DMSO) and 2 (prepared with DMSO) show a different behaviour. RC film 1 was destroyed at a pressure of 2.5 bar corresponding to the maximum flow rate 4.5 ml/min. Meanwhile, RC film 2 was broken at a pressure of 1.4 bar corresponding to the maximum flow rate 2.5 ml/min. This forced us to investigate the long-term stability even at different pH. Thus, the films were immersed in acid media (pH = 3) and basic media (pH = 10) for 2 days and measured. The hydraulic pressure is comparable with the fresh RC films, showing a high storage stability in these media. The new results and explanations are inserted as Figure 5B and on page 12, line 253 and page 13, line 260-274.

All these discussions revealed a strong dependence on the presence of DMSO during the coagulation process. We have inserted a detailed discussion on the role of DMSO on page 14, line 285-290.

As mentioned by the referee the tensile strength/elongation is also very important, however, mainly for applications such as films or foils for packaging. We are sorry, but due to limitation in equipment we were not able to investigate this property. It will be part of a future cooperation.

2. P13, L262...

(PEG) as molecular probes (Figure 6B).

I think Figure 5B is correct.

Comment: We thank the reviewer for the comment and apologize for the confusion. We have now corrected this to Figure 5C (P13, L276).

3. P14, L282

The authors mentioned that “these results suggest a negative charge surface of the cellulose.” I understand the results, but why? The OH-group of cellulose is not modified with any chemicals in this process, right? Please add the explanations in the manuscript.

Comment: We thank the reviewer for the comment. Indeed, the surface is not chemically modified. However, the use of electrolytes may induce a change in the surface charge due to sorption of anions or ionization by cations. We discussed this in more detail on P15, L304-306.

4. P15, L301

The [TBP]⁺ anion is not...

I think cation is correct.

Comment: We thank the reviewer again for the correction. We have fixed this (P16, L323) and proofread the manuscript again. Different minor corrections are highlighted in the text.

5. P19, L399

Please add the apparatus information.

Comment: We excuse the limited information. We have added the information about the TGA measurement (P20, L421 – L424).

Reviewer #2 (Remarks to the Author):

Development of sustainable route for the clean production of regenerated cellulose is of vital importance for the wider application of cellulose and sustainable development of human society. In this work, the authors used TBPH solution to dissolve MCC and get regenerated cellulose by adding organic carbonates. Some properties of the regenerated cellulose film were characterized. The results obtained are quite interesting, but the following points have to be fully addressed before acceptance. So major revision is needed.

Detailed comments:

1. The title should be modified. At least, TBPH solution should be mentioned.

Comment: We agree with the comment of the reviewer. Indeed, for a better visibility in terms of web search we modified the title "A New Sustainable Route Towards Regenerated Cellulose Films using Tetrabutylphosphonium Hydroxide and Organic Carbonates"

2. I think the properties of the regenerated cellulose film obtained from this work should be compared with the regenerated cellulose film obtained by Viscose process or other reported processes based on tests or literature results, for instance, mechanical strength, barrier properties?

Comment: We acknowledge the reviewer for the constructive suggestion on the paper. It is hard to compare the RC films synthesized via our new method and the Viscose process. The properties always are depending on the following applications. With the Viscose process, hard materials of solidified cellulose were obtained having a tensile strength of 135 MPa. Here we obtain a soft/swollen film with low stability. But as mentioned for the referee 1 in the current state we are not able to investigate this property due to limitation in equipment. But it will be part of a future cooperation since we are interested to improve the stability of the films.

As mentioned for referee 1 we focused on an application in filtration technology. Therefore, we investigated the mechanical durability/stability based on the hydraulic pressure of the flow through the film. The new results and explanations are inserted as Figure 5B and on page 12 and 13.

3. Page 12, line 231, please indicate how the thickness of the films decreased to around 200 nm? By drying?

Comment: We thank the reviewer for the comment. We apologize for a mistake in the thickness. The thickness of the film reduced to 200 μm (not as stated nm). Thus it has nearly the same high as the casting high of 500 μm . We believe the reduction in thickness was probably caused by the coagulation process and partially drying during the LM preparation process. We have added a comment on page 11, line 229 – page, 12, line 234.

4. Page 19, line 400, please indicate the gas flow speed of nitrogen.

Comment: We excuse the limiting information. We have added the information about the TGA measurement (Page 20, line 421 – line 424).

5. Please indicate the test mode of AFM.

Comment: In the Methods section, we have added the following information about the AFM measurement (Page 21, line 429 – line 433): "In order to avoid excessive sample, contact the device was operated in the so-called non-contact mode with small cantilever oscillation amplitudes of a few nm and moderate setpoint (~70% of free amplitude) while the dither frequency was kept constant (amplitude modulation). Multiple sample locations were probed to ensure validity of obtained morphology."

REVIEWERS' COMMENTS:

Reviewer #1 (Remarks to the Author):

The manuscript was revised correctly according to the reviewer's comments. The revised manuscript is suitable for publication.

Reviewer #2 (Remarks to the Author):

The authors have addressed all points that raised by reviewers. The current version of this manuscript can be accepted.

Gaaino PDF Trial
www.gaaino.com

Response to Reviewers

Reviewer #1 (Remarks to the Author):

The manuscript was revised correctly according to the reviewer's comments. The revised manuscript is suitable for publication.

Reviewer #2 (Remarks to the Author):

The authors have addressed all points that raised by reviewers. The current version of this manuscript can be accepted.

Comment: We are glad that both reviewers agree with our statements.